# Physicochemical Evidence that *Francisella* FupA and FupB Proteins Are Porins

**DOI:** 10.3390/ijms21155496

**Published:** 2020-07-31

**Authors:** Claire Siebert, Corinne Mercier, Donald K. Martin, Patricia Renesto, Beatrice Schaack

**Affiliations:** 1TheRex Team, TIMC-IMAG, CNRS, INP, Université Grenoble Alpes, 38000 Grenoble, France; siebert.claire@gmail.com (C.S.); patricia.renesto@univ-grenoble-alpes.fr (P.R.); 2GEM Team, TIMC-IMAG, CNRS, INP, Université Grenoble Alpes, 38000 Grenoble, France; corinne.mercier@univ-grenoble-alpes.fr; 3SyNaBi Team, TIMC-IMAG, CNRS, INP, Université Grenoble Alpes, 38000 Grenoble, France; don.martin@univ-grenoble-alpes.fr; 4CEA, IBS, CNRS, Université Grenoble Alpes, 38044 Grenoble, France

**Keywords:** *Francisella tularensis*, FupA, FupB, porins, fluorescence flux, impedance spectroscopy

## Abstract

Responsible for tularemia, *Francisella tularensis* bacteria are highly infectious Gram-negative, category A bioterrorism agents. The molecular mechanisms for their virulence and resistance to antibiotics remain largely unknown. FupA (Fer Utilization Protein), a protein mediating high-affinity transport of ferrous iron across the outer membrane, is associated with both. Recent studies demonstrated that *fupA* deletion contributed to lower *F. tularensis* susceptibility towards fluoroquinolones, by increasing the production of outer membrane vesicles. Although the paralogous FupB protein lacks such activity, iron transport capacity and a role in membrane stability were reported for the FupA/B chimera, a protein found in some *F. tularensis* strains, including the live vaccine strain (LVS). To investigate the mode of action of these proteins, we purified recombinant FupA, FupB and FupA/B proteins expressed in *Escherichia coli* and incorporated them into mixed lipid bilayers. We examined the porin-forming activity of the FupA/B proteoliposomes using a fluorescent 8-aminonaphthalene-1,3,6-trisulfonic acid, disodium salt (ANTS) probe. Using electrophysiology on tethered bilayer lipid membranes, we confirmed that the FupA/B fusion protein exhibits pore-forming activity with large ionic conductance, a property shared with both FupA and FupB. This demonstration opens up new avenues for identifying functional genes, and novel therapeutic strategies against *F. tularensis* infections.

## 1. Introduction 

The facultative intracellular Gram-negative coccobacillus *Francisella tularensis* is the etiologic agent of tularemia [1]. The only two subspecies of *F. tularensis* that cause severe disease in humans are the subsp. *tularensis* (Type A strains) and subsp. *holarctica* (Type B strains). Because of its extremely high infectivity (<10 bacteria are sufficient to induce severe infection) and its potential use as a biological weapon, the aerosolizable and high-mortality rate *Francisella* pathogen has been classified as a class A bioterrorism agent by the U.S.A. Centers for Disease Control and Prevention (CDC).

No safe and potent vaccine is currently available to prevent infection by *Francisella* [1]. In the mid-1990s, Russian scientists developed a vaccine based on an attenuated mutant selected from a virulent isolate of *F. tularensis* subsp. *holarctica* [2]. This *F. tularensis* LVS was used for protection of at-risk laboratory staff. However, because this strain was lethal in mice and its reversion to virulence could not be excluded, it remained unlicensed in the USA and the European Union [3]. A later study which aimed at the creation of an attenuated Type A mutant strain led to the discovery of a spontaneous *F. tularensis* subsp. *tularensis* SCHU S4 mutant, which was designated FSC043. It exhibited an improved efficacy against both systemic and aerosol challenges in mice [4]. Proteomic analysis of this mutant identified protein candidates that could be responsible for the reduction in virulence. Among these candidates were the paralogous proteins encoded by FTT0918 and FTT0919. Neither protein was expressed in the FSC043 strain, which instead codes for a hybrid protein that consists of the FTT0918 N-terminal and the FTT0919 C-terminal domains. Interestingly, such a fusion protein, which is believed to originate from a genomic deletion resulting from a recombination event between the two paralogous genes, was also observed in the LVS strain [5]. In contrast to the deletion of the gene encoding FTT0919 that had no effect on bacterial virulence in a mouse infection model, inactivation of the gene encoding FTT0918 was found to contribute significantly to the attenuation of both the SCHU S4 [4] and the LVS strains [5]. The name FupA (Fer Utilization Protein A) was proposed for this 58 kDa protein since it was very similar to the *F. tularensis* siderophore receptor FslE that was required for the acquisition of siderophore-bound iron [6]. The name FupB was given to the protein encoded by the adjacent FTT0919 paralogous gene for which no relationships with metal acquisition or bacterial virulence had been shown [4,7]. The LVS chimeric protein (locus FTL0439) implicated in iron metabolism was termed FupA/B [8].

We recently performed a comprehensive phylogenetic analysis of the FupA and FupB homologous proteins [9]. These outer membrane proteins (OMPs) comprise the five subfamilies FupA, FupB, FslE, FmvA, and FmvB, which share a domain of unknown function (DUF3573) and display specific features that might reflect functional differences. To date, very little is known about the functions of these proteins. From the observation that FupA expression is regulated neither by the ferric uptake regulator (Fur) protein nor by the iron level, it was suggested that this protein may not be involved in iron acquisition and that it could instead be involved in the transport of additional substrates [10]. Among such unexpected functions, we demonstrated that the deletion of *fupA* or *fupA/B* contributes to reduced antibiotic susceptibility and could promote the emergence of antibiotic resistance mediated by an increased biofilm formation [9]. This effect was related to the hyper-production of vesicles by the *fupA* deletion mutant and correlates well with the presence of a lipoprotein motif at the N-terminus of FupA, which deletion would compromise the membrane stability. The functional role of FupB remains to be clarified since, in contrast to what had been described for *fupA* or *fupA/B*, removal of *fupB* failed to alter iron metabolism [7], bacterial virulence [4], or antibiotic resistance [9].

Although FupA and FupB are identified as orthologous proteins using the BLASTP software, their functions are not preserved [9,11]. The evolution of gene-phenotype relationships is complex, and functional differences between orthologues are, in some cases, greater than expected [12]. Due to the absence of structural data available in the proteomic databases, we first performed structural modeling using protein structure prediction programs in order to gain some insight into the functional roles of both FupA and FupB. Bioinformatic simulations provided evidence that, as previously predicted for FupA using the Hidden Markov Model-based PRED-TMBB program [10], both FupB and FupA/B fold partially as β-barrels in the outer membrane of *Francisella*. FupA and FupA/B have been reported to facilitate iron transport [10,13,14].

From these data, we hypothesized that these three *Francisella* OMPs, which are partially folded as β-barrels, assemble to form channel-forming transmembrane porins. Such a porin could allow the influx of essential nutrients or contribute to the envelope stability of Gram-negative bacteria [15]. Then, using fluorescent flux assays and electrophysiology, we demonstrated that FupA, FupB, as well as FupA/B assemble as functionally active porins when correctly folded in a lipid environment.

## 2. Results and Discussion

### 2.1. Structural Modelling of Fup Proteins Predicted Their Folding as Porins

FupA and FupB are paralogous proteins that consist in 558 and 482 amino acids, respectively. Alignment of both proteins with the 552 amino acid sequence of FupA/B has already been reported [9]. The amino acids 1 to 307 of FupA/B are 98.7% identical to the N-terminal region of FupA, while its C-terminal half (amino acids 298 to 552) is strictly identical to the C-terminal part of FupB. The three-dimensional (3D) structures of the three proteins were predicted using the Phyre 2 server [16]. The automatic alignments of the C terminal regions of FupA and FupA/B (56% of their sequence) and FupB (63% of its sequence) with available structures were modeled with respectively 98.3% (FupA), 98.9% (FupB), and 98.8% (FupA/B) accuracies by the single highest scoring template. As expected when considering amino acid sequence homologies, the three partial predicted structures (Figure 1) were very similar to each other and showed clear homologies with known porins mainly folded as β-barrels, in particular with the outer membrane protein OprP from *Pseudomonas aeruginosa* [17]. The obtained model (Figure 1) depicted the *Francisella* Fup proteins as monomers spanning the membranes by the arrangement of a 16-stranded β-barrel. As for OprP, FupA, FupB, and FupA/B are expected to form trimers. 

### 2.2. Fluorescence Assays on Proteoliposomes Containing FupA/B Suggested Porin Activity

Insertion of the chimeric FupA/B protein into liposomes was used as a means to evaluate the insertion capability of the FupA protein family into membranes and their permeability to a fluorescent molecule. The integration of FupA/B into liposomes was analyzed through a discontinuous sucrose gradient. Dot blot analysis of the gradient fractions with anti-His antibodies failed to reveal the presence of FupA/B in the lowest fraction of the gradient (40% sucrose), indicating that the protein did not precipitate during the integration of the protein into the liposomes (Figure 2B). A thin white band associated with the formed PLs could be observed at the interface between the 0 and the 10% sucrose cushions (Figure 2B).

The number of FupA/B proteins per PL was calculated in several steps, as follows.
The intensity of the FupA/B light signal emitted by the PL dot was quantified as being 73% of that emitted by the dot corresponding to the starting material (FupA/B-MBP-His tagged protein at 100 μg/mL) (Figure 2B). Given that the sucrose gradient fractions were of 1 mL, we estimated that the whole PL fraction contained 73 μg of FupA/B protein. Knowing that the molar mass of MBP-FupA/B is 100 kDa (Reference [9] and Figure 2A), we estimated that there were 73 × 10^−11^ moles of FupA/B, i.e., 4.38 × 10^14^ molecules, of FupA/B inserted in the total number of PLs.We calculated the number of lipid molecules per PL by dividing the whole surface occupied by lipids in one PL by the mean surface of one phospholipid. Dynamic Light Scattering (DLS) measured the mean radius of PLs as being 63.5 ± 3.6 nm (mean ± SD; *n* = 3). Given that PLs contain 2 layers of lipids, we calculated the whole surface occupied by lipids in one PL as 2 × (4π R^2^), thus as 1.01 × 10^5^ nm^2^. Knowing that the mean surface of a phospholipid is estimated to be 0.65 nm^2^ [18], we estimated the number of lipid molecules per PL as being 1.01 × 10^5^/0.65 = 1.55 × 10^5^.To calculate the number of formed PLs, we divided the total number of lipid molecules found in all the PLs by the number of lipid molecules per PL. The liposomes were prepared with 2 mg, thus 2.85 µM, i.e., 1.71 × 10^18^ molecules of lipids. Assuming that there was no loss of lipids between the liposome formation step and that of PLs, the number of formed PLs was thus estimated to be 1.71 × 10^18^/1.55 × 10^5^ = 1.03 × 10^13^.Finally, the number of FupA/B monomers per PL was calculated as the total number of FupA/B proteins in the whole number of PLs, divided by the estimated number of PLs, thus 4.38 × 10^14^/1.03 × 10^13^ = 39.7.


This number of protein monomers per PL was in the range of the typical membrane protein occupation range in PLs obtained using detergents [19]. 

To assess the potential porin activity of FupA/B, we then measured the efficacy of entrance of the ANTS dye into the PLs. Gel filtration through a Sephadex column had been reported as the method of choice for separating liposome-encapsulated molecules from those remaining outside the liposome membranes [20,21]. Fluorescence measurement of the fraction collected from the G25 column, on which empty liposomes pre-incubated with ANTS had been loaded, revealed values close to 0. This indicated that the membrane of empty liposomes was not permeable to the ANTS dye and that filtration through the G25 column efficiently excluded ANTS from the liposome solution (Figure 2C, “empty liposomes” column). In contrast, measurement of the fluorescence of the fraction collected from the G25 column loaded with a mixture of PLs pre-incubated with ANTS returned an ANTS concentration of 3.70 ± 0.01 μM (Figure 2C, “FupA/B PL” column). Altogether, these results demonstrated that ANTS could penetrate FupA/B PL and that FupA/B is a porin that allows fluorescent molecules such as ANTS to pass through a membrane bilayer.

To calculate the amount of ANTS which had entered the PLs through the FupA/B porin, we proceeded again by steps, as follows:
Knowing that the PLs’ radius estimated by DLS was 63.5 ± 3.6 nm, we calculated the volume of any given PL as 4/3 × π × R^3^ = 1.072 × 10^−21^ m^3^ = 1.072 × 10^−18^ L.Knowing that 2 mg of lipids had been used for the experiment, with the production of 1.013 × 10^13^ PLs, the total volume occupied by the formed PLs in the 1 mL working fraction was calculated as 1.013 × 10^13^ × 1.072 × 10^−18^ = 1.04 × 10^−5^ L = 11.04 μL. The total volume of PLs thus represented 1.10% of the 1 mL working fraction.Given that PLs were incubated in a 20 mM ANTS solution and that the total volume of PLs represented 1.10% of the total volume (1 mL), we reasoned that the maximum concentration of ANTS reached inside the PLs would have been 20 × 10^3^ × 1.10% = 220 μM if their membrane had been permeable to the dye. In contrast, we obtained a concentration of 3.70 ± 0.01 μM ANTS within the PLs (Figure 2C), which represented only 1.7% of the ANTS molecules that would have diffused passively through a permeable membrane.Given the ANTS concentration calculated within the PLs (3.70 ± 0.01 µM) and the calculated volume of one PL (1.072 × 10^−18^ L), we calculated that each PL would contain 4.02 × 10^−24^ moles of ANTS, i.e., 2.42 molecules of ANTS, while it would have contained 142 molecules if the PL membrane had been permeable to ANTS.


This calculation clearly demonstrated that ANTS molecules did not diffuse passively through the empty PL membrane and that their entry into the PL required instead their fluorescent flux through the FupA/B proteins. This property strongly suggested a FupA/B porin activity.

### 2.3. Impedance Spectroscopy Experiments Using Tethered Lipid Bilayer Membranes Confirmed the Porin Activity of FupA/B

We have already shown in a previous study that impedance spectroscopy provides a precise measurement of the ion flux driven by the OprF porin across membranes [22]. To evaluate the porin activity of FupA, FupB, and FupA/B proteins, we thus used tethered lipid bilayer membranes (t-LBMs) generated by a TethaPod device (SDX Tethered Membranes, Australia). We first measured the steady-state conductance of this t-LBM system using impedance spectroscopy controlled by the TethaPod, in the presence of recombinant FupA/B vs buffer A (control experiment) added to one side of the t-LBM. The insertion of FupA/B into the t-LBM led to a reproducible and steady-state conductance that stabilized at 0.2–0.25 µS within a few minutes (Figure 3A). Specifically, we verified that the capacitance varied only slightly (18%) upon the incorporation of FupA/B to the t-LBM (Figure 3B). According to the manufacturer’s recommendations, an almost constant capacitance of ~10 nF upon addition of the protein to the t-LBM indicated that the thickness of the membrane did not vary through the experiment and thus excluded the possibility of a lipid multilayer, which would have been inappropriate for protein insertion into the membrane.

To characterize the ion permeability of the FupA/B porin, we added increasing NaCl concentrations into the upper part of the t-LBM wells. From Figure 3C, we concluded that the porin FupA/B was permeable to NaCl. In the control well (no protein) we noticed a small increase in conductance for concentrations of NaCl greater than 500 mM, which suggested that only a small increase in permeability of the t-LBM resulted from the very large osmotic difference across the membrane (in blue in Figure 3C). 

Following conductance measurements, the proteins potentially inserted in the t-LBM were then solubilized by the addition of an SDS solution into the wells followed by their separation by SDS-PAGE and immunoblotting using antibodies specific to FupA. A band migrating at the apparent molecular weight expected for FupA/B confirmed the correct insertion of the protein in the t-LBM (Figure 3D, T lane). The size of FupA/B protein in lane T (~60 kDa, Figure 3D) is ~40 kDa less than that of the recombinant FupA/B detected by Coomassie blue staining of the SDS-PAGE shown in Figure 2A due to the TEV protease digestion. We concluded that the increase in conductance detected in Figure 3A was due to the porin activity of FupA/B inserted in the t-LBM. 

Since FupA/B, FupA, and FupB are orthologous proteins, we tested the three proteins for their potential porin activities on the same Tethapod chip. In Figure 4A, we verified after washes of the well that the resulting conductance of FupA or FupB insertion in the t-LBM membrane was similar (0.2 µS) to the one obtained in Figure 3A for FupA/B. The washes of the well of the TethaPod did again produce a loss of conductance for FupA/B. The final conductances of FupA and FupB were around 0.25 and 0.3 µS, respectively (Figure 4B). The conductances of the 3 tested Fup proteins with or without their MBP tag were very similar. These high conductances suggested that the insertion of these proteins into t-LBM was effective. Their insertion in the t-LBM was as fast as the one of FupA/B. These experiments were done thrice on 3 different chips and we always found slightly more conductance with FupB than with FupA (around a 1.25-fold increase) (Figure 4B).

Finally, solubilization of the proteins present in the t-LBM by SDS followed by their separation by SDS-PAGE and immunoblotting using antibodies specific to FupA confirmed the insertion of FupA in the t-LBM and the apparent molecular mass detected for FupA was in agreement with its cleavage by the TEV protease (Figure 4C). Altogether, these experiments let us to conclude that FupA and FupB are porins.

## 3. Material and Methods

### 3.1. Protein Purification

The High-Fidelity PCR master mix (Phusion, Finnzymes, New England Biolabs, Evry, France) was used to amplify both the *fupA* and the *fupB* coding sequences from the *F. novicida* U112 strain (CIP56.12-Centre de Ressources Biologiques de l’Institut Pasteur, Paris, France) genomic DNA. The PCR product encoding FupA/B used the genomic DNA of the *F. tularensis* LVS strain (NCTC 10857) as a template. PCR primers were designed to amplify the coding sequence, excluding the signal peptide. The obtained amplicons were cloned into the pDON201 entry vector, then transferred into the pETG-41A vector, which contains an N-terminal 6xHis-tag followed by a maltose binding protein (MBP) tag, using the Gateway cloning system (Invitrogen, Villebon-sur-Yvette, France)) according to the manufacturer’s instructions. The Tobacco Etch Virus (TEV) cleavage site is located between the MBP- and the cDNA sequences, allowing subsequent removal of both tag sequences. Construct integrity was verified by DNA sequencing (Eurofins, Genomics, Ebersberg, Germany). All the primers used in this study are listed in Appendix A.

The production of recombinant proteins was performed in the engineered *E. coli* strain BL21 (DE3) grown at 37 °C, under 150 rpm, in Luria Broth (LB) supplemented with 50 µg/mL kanamycin. The same protocol was used for the production of FupA, FupB, and FupA/B. When the OD_600nm_ of the bacterial cell suspension reached 0.5, protein expression was induced by the addition of 0.5 mM isopropyl-β-D-thiogalactopyranoside (IPTG). Following overnight incubation at 16 °C, bacteria were pelleted by a 20 min centrifugation at 5000× *g*, solubilized in lysis buffer (50 mM Tris pH 8.8, 200 mM NaCl, 10 mM imidazole, 1% CHAPS and Complete Protease Inhibitor^®^ (Roche Diagnostics, Meylan, France), then disrupted by sonication. Following centrifugation (60,000× *g*, 30 min, 4 °C), the bacterial lysate was loaded onto a Ni^2+^-NTA column (Qiagen, Courtaboeuf Cedex, France) for affinity purification. Following extensive washes of the column (50 mM Tris pH 8.8, 200 mM NaCl, supplemented sequentially with 25, then 50 mM imidazole) the proteins were eluted with 300 mM imidazole. The fractions containing the purified proteins, as assessed by SDS-PAGE and Coomassie blue staining (Figure 2A), were pooled and dialyzed overnight at 4 °C against 50 mM Tris pH 7.5, 200 mM KCl (buffer A). Following dialysis, the protein was concentrated using an Amicon Ultra centrifugal device equipped with a 10 kDa cut-off membrane and further stored at −80 °C after flash-freezing.

### 3.2. Protein Insertion in Liposomes

Insertion of the recombinant proteins into liposomes was achieved as previously described [18] according to Geertsma et al. [23]. Pure lipids (1,2-dioleoyl-sn-glycero-3-phosphoethanolamine or PE, 1,2-dioleoyl-sn-glycero-3-phosphoglycerol or PG and 1′,3′-bis[1,2-dioleoyl-sn-glycero-3-phospho]-sn-glycerol = cardiolipin or CL) were purchased as chloroform solutions (Avanti Polar Lipids, Alabaster, AL, USA). Liposomes composed of PE:PG:CL (4:4:2 molar ratio) were extruded first through a 200 nm, then a 100 nm polycarbonate porous membrane to produce large unilamellar vesicles (LUVs) that were diluted to the final concentration of 2 mg/mL. Unless otherwise stated, the buffer A in which the proteins were dialyzed was also used for all further dilutions. The quality of the liposomes was evaluated using dynamic light scattering (DLS) (radius: 70.3 ± 3.5 nm, polydispersity: 31 ± 9.5% (mean ± SD; *n* = 3)). Destabilization of liposomes (2 mg in buffer A) was induced by the addition of Triton X-100 (Sigma Aldrich, Saint-Louis, MO, USA) and monitored using measurement of the absorbance at 550 nm (A_550_) [18,23,24,25]. The use of Triton X100 was recommended by Geertsma et al. [23] as a way to reduce the time needed to equilibrate preformed liposomes compared to milder detergents. This property is essential for the insertion of many membrane proteins in liposome membranes. In addition, for most membrane proteins, this reconstitution was also proven [23] to be most efficient when using detergent concentrations slightly above the saturation point, Rsat, which refers to the Triton X-100 concentration required to saturate the membrane and for which the turbidity of the suspension evaluated by A_550_ reaches its maximum [23,24]. In our experiments, the titration of 1.425 mL of preformed liposomes was performed with an increasing concentration in Triton X-100 and Rsat was estimated to be 0.1 mM. 2 mg of liposomes were dispersed into 1.425 mL of buffer A supplemented with 0.3 mM Triton X-100 and 97.5 µg of purified recombinant FupA/B still containing its MBP solubility tag (75 µL of recombinant protein at 1.3 mg/mL in buffer A). The FupA/B-liposomes-detergent mixture was incubated for 1 h at 25 °C on a rotating wheel and in a final volume of 1.5 mL. As suggested in the Geerstma’s article [23], the detergent was removed by 4 successive incubations of 30 min each, at different temperatures (25 °C to ensure maximal agitation, then 3 incubations at 4 °C to keep the protein intact) with fresh Biobeads SM-2 (Biorad, Marnes-la-Coquette, France) in large 2 mL tubes and using a Biobeads/detergent ratio of 70/1 (*w*/*w*), according to the manufacturer’s instructions, in order to prevent protein precipitation. To separate the PL fraction from the precipitated proteins, the FupA/B-liposomes mixture was loaded onto the top of a 4-step discontinuous sucrose gradient (i.e., 40%, 30%, 20%, and 10% sucrose concentrations) prepared in 11 mL of buffer A. After ultracentrifugation (280,000× *g*, 2 h, 4 °C) in a swinging rotor (Thermo Scientific Sorvall, ThermoFisher Scientific, Bourgoin-Jallieu, France), eleven 1 mL fractions were recovered. To characterize the PLs, 2 µL of each fraction were spotted onto a nitrocellulose membrane, which was incubated for 60 min at room temperature with anti-6xHis antibodies coupled to horseradish peroxidase (HRP, Sigma Aldrich, St. Louis, MO, USA) (dilution: 1/10,000). The dot-blot was then washed twice for 10 min in TBST (Tris-Buffered Saline containing 0.5% Tween-20) and incubated for 1 min in ECL substrate chemiluminescent detection reagent (Biorad, Marnes-la-Coquette, France). The chemiluminescent dot-blot was imaged using a ChemiDoc MP imager (Bio-Rad, Hercules, CA, USA) and the intensity of the light signal emitted by each dot was quantified in relation to the signal emitted by 2 μL of recombinant FupA/B (at 100 μg/mL in buffer A) spotted on the same membrane. The quantification was performed using the ImageLab software (Biorad, Hercules, CA, USA). The resulting PL fraction was stored at 4 °C until being used.

Quantification of the lipids present in the PLs was based on the integration of fluorescent 8-anilino-1-naphthalene sulfonic acid molecules (ANS, Sigma, St. Louis, MO, USA) into their membranes, as previously described [18]. Briefly, a calibration curve was set using the measurement of the fluorescence emitted by empty LUVs aliquots (0 to 0.5 mg/mL) incubated in a 7.66 × 10^−4^% ANS solution (excitation wavelength 310 ± 8 nm, emission wavelength 460 ± 8 nm). This calibration curve was used to calculate the quantity of lipids present in PL aliquots based on their emitted fluorescence when incubated with ANS (same concentration as used for the samples constituting the calibration curve).

### 3.3. Porin Activity by Fluorescent Uptake Assay in Proteo-Liposomes

Porin activity was evaluated by comparing the entry of fluorescent molecules into PLs vs empty liposomes, as described by Stockbridge et al. [25]. Also, 500 µL of FupA/B PLs (containing 1 mg of lipids) were incubated for 3.5 h with 20 mM fluorescent 8-aminonaphthalene-1,3,6-trisulfonic acid, disodium salt (ANTS, ThermoFisher, Bourgoin-Jallieu, France) [26]. This molecular reagent is a polyanionic dye often used to measure membrane permeability. During the incubation, the dye penetrates the PLs only through pores formed in the membrane. The mixture (dye + PLs) was then loaded onto a G25 desalting column (GE Healthcare, Chicago, IL, USA) equilibrated in buffer A to retain on the column the excess of ANTS molecules that had not entered the PLs. The eluted fraction contained the PLs charged with ANTS molecules that had entered the PLs through the FupA/B proteins inserted in the membranes. The fluorescence emitted by ANTS-charged PLs was compared to that emitted by a similar amount of empty liposomes previously incubated with ANTS in the same conditions as those used for PLs, and using black 96-well Greiner microtiter plates (Sigma Aldrich, St. Louis, MO, USA) and a Varioskan LUX Multimode Microplate Reader (excitation wavelength: 405 nm; emission wavelength: 535 nm; bandwidth: 12 nm; *n* = 2). The correlation between ANTS fluorescence and ANTS concentration was linear between 0 and 4 µM. It is important to note that, in the G25 elution fractions, ANTS molecules would equilibrate their concentration between the inner volume of the PLs and the external milieu, using the inserted FupA/B proteins to cross the membrane.

### 3.4. Electrophysiological Measurement of Porin Ion Conductance in Tethered Lipid Bilayer Membranes

To evaluate the porin activity of FupA, FupB, and FupA/B proteins, we used a commercially available device, i.e., a TethaPod (SDX Tethered Membranes, Roseville, Australia) for the production of t-LBMs. The device comprised 6 wells that contained a pre-formed tethering layer made of benzyl disulfide undecaethylene glycol phytanol and benzyl disulfide tetra ethylene glycol polar spacer molecules in a 10:90 ratio. A t-LBM was formed in each of the 6 wells by addition of a 3 mM ethanolic solution of glycerodiphytanylether: diphytanylether phosphatidylcholine (30:70 molar ratio) (AM199, SDX Tethered Membranes), which self-assembled into a lipid bilayer membrane after being rinsed with buffer A. The porins were incorporated into the t-LBMs after the addition into each well of 15 µL of a PL mixture that contained either the recombinant FupA, FupB, or FupA/B (each protein at 0.4 mg/mL concentration in buffer A). In the wells, the His-MBP tag was cleaved off the proteins by digestion for 10 min with 1.5 µg of TEV-EDTA. The proteins without the His-MBP tag were not soluble anymore and preferentially incorporated in the t-BLM membrane. The negative control consisted in the same amount of buffer A without any protein. To measure the conductance of pore-forming proteins, the TethaPod system used an alternating-current (AC) impedance spectroscopy technique that was adapted to t-LBMs [27]. To evaluate Na^+^ transport through the porin, increasing concentrations of NaCl (0 to 1 M in buffer A) were superfused into each well. We chose to vary NaCl since KCl was already in buffer A (thus on both sides of the t-LBM). The TethaPod applied a sequential 20 mV excitation over the 1 kHz–0.1 Hz frequency range. The output measured from the electrical impedance spectroscopy (EIS) was modeled using the Tethaquick software (v2.0.49, SDX Tethered Membranes, Roseville, Australia) to determine the capacitance (C_m_) of the t-LBM, its resistance (R_m_), and a constant phase element accounting for the impedance of the tethered region below the t-LBM (CPE_teth_). During the experiments, each t-LBM remained stable, as shown by the low variability of the t-LBM capacitance before porin incorporation (8.3 ± 0.1 nF, mean ± SD, *n* = 15).

### 3.5. Evaluation of the Amount of Protein Inserted in the Tethered Lipid Bilayer Membrane

To visualize the proteins inserted in the t-LBM, the wells of the TethaPod system were firstly washed thoroughly with buffer A (filling up the wells with buffer A, then emptying them, and repeating these actions 10 times). Then, 15 µL of Coomassie loading-buffer containing SDS (Biorad, Hercules, CA, USA) were added into each empty well in order to solubilize the lipid bilayer and its inserted proteins. The protein content of each well was run through an SDS-PAGE (12% polyacrylamide) and transferred to a nitrocellulose membrane. The FupA and FupA/B proteins were revealed using rabbit anti-FupA antibodies (dilution: 1:10,000) [9], followed by HRP-conjugated anti-rabbit antibodies (Sigma-Aldrich, St. Louis, MO, USA) diluted 1 to 10,000 in TBST buffer, 5% nonfat milk. Following two 10-min washes in TBST at room temperature, the membrane was incubated for 1 min in ECL substrate chemiluminescent detection reagent (Bio-Rad, Hercules, CA, USA) and imaged with the ChemiDoc MP imager (Bio-Rad, Hercules, CA, USA).

### 3.6. Statistical Analyses

Statistical analyses were performed using one-way ANOVA (GraphPad Prism software, GraphPad company, San Diego, CA, USA).

## 4. Conclusions

To the best of our knowledge, the involvement of porins in ferrous iron acquisition by *F. tularensis* had been hypothesized, but never firmly established. In this study, we used recombinant His-MBP-tagged proteins and an optimized relipidation protocol to successfully produce FupA/B proteoliposomes that were characterized at a functional level using the ANTS fluorescent probe. The fluorescent substrate ANTS was taken up by proteoliposomes, suggesting porin activity. We confirmed the porin activity by measuring the ion conductance using impedance spectroscopy. The measurements reported here provide evidence that FupA, FupB, and FupA/B are porins when inserted into tethered lipid bilayer membranes.

These conclusions were supported by a structural modeling approach using Phyre2 that predicted a β-barrel structure involving 56–63% of the FupA, FupB, FupA/B protein sequences with structural similarity with the OprP protein from *P. aeruginosa*. This is in agreement with a previous report that predicted that FslE and FupA would be folded as β-barrels in the outer membrane with periplasmic plug domains similar to TonB-dependent receptors. In addition, the deletion mutants of *fupA*, *feoB*, or *fslE,* and which proteins are involved in siderophore-dependent iron transport, were shown to be viable and virulent, suggesting the existence of compensatory systems to support growth and virulence. This similarity suggests that the three proteins FupA, FeoB, and Fsl function in a similar manner. Formation of complex proteoliposomes containing the three proteins in their membrane coupled to the use of fluorescent probes should help model that FupA functions as a high-affinity transport porin for copper, as it does for iron, with both nutrients being normally at limiting levels in the host environment. 

## Figures and Tables

**Figure 1 ijms-21-05496-f001:**
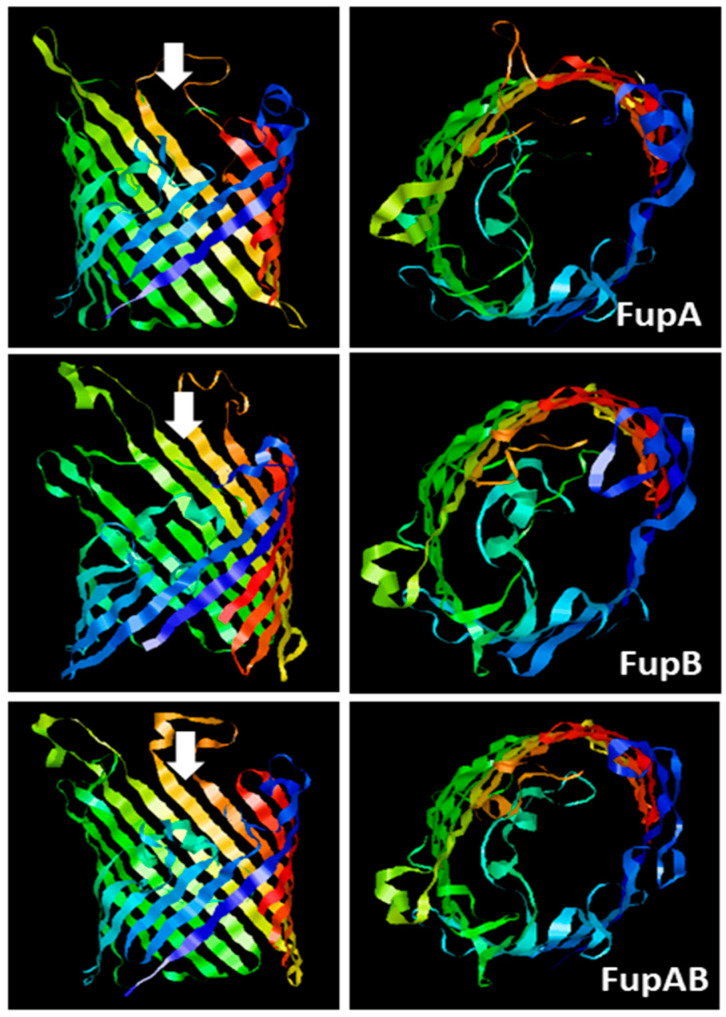
Predicted 3D structures of *F. tularensis* FupA, FupB, and FupA/B monomers. The C terminal regions of the 3 proteins (310 amino acids from C terminal part of FupA; 304 amino acids C terminal part of FupA/B and FupB) were analyzed using the Phyre2 server. The panels on the left show a lateral view of each transmembrane region constituted of 16-barrels, while the panels on the right show a top view of each transmembrane region, observed from the intracellular compartment and in the direction of the white arrows.

**Figure 2 ijms-21-05496-f002:**
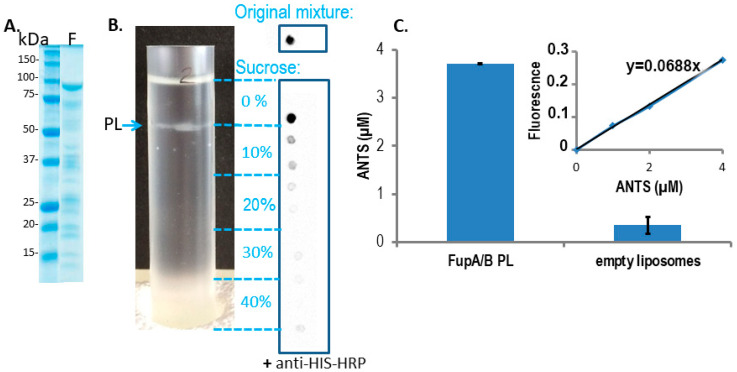
FupA/B inserts spontaneously into liposome lipid bilayers. (**A**) Coomassie blue staining gel of the purified MBP-FupA/B protein. Marker molecular weights are indicated (kDa). (**B**) The sucrose density gradient (0 to 40%) (left panel) and dot blot analysis of FupA/B (right panel) revealed that the PLs concentrated at the interface between the 0% and the 10% sucrose steps. 2 µL of each sucrose gradient fraction were spotted on a nitrocellulose membrane revealed by anti-His antibodies conjugated to horseradish peroxidase (HRP) and chemiluminescence. The top panel shows the signal obtained with 2 μg of recombinant FupA/B. (**C**) Comparison of the ANTS fluorescence signals (± SD) from 100 µL of FupA/B PL vs 100 µL of liposomes after G25 filtration (Ex: 405 nm; Em: 535 nm; *p*-value = 0.0014 between both histograms, non-parametric one-way ANOVA). A linear correlation between the ANTS concentration and the emitted fluorescence was verified via the titration of pure ANTS. These data are representative of 3 independent experiments performed in duplicate and providing similar results.

**Figure 3 ijms-21-05496-f003:**
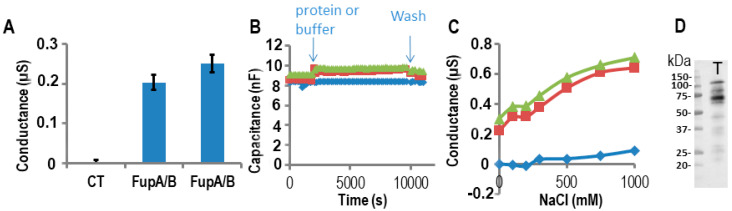
Measurements of FupA/B porin activity using impedance spectrometry on a TethaPod chip. FupA/B was digested by the TEV enzyme in the well. (**A**) The conductance of the t-LBM after the insertion of FupA/B (2 independent wells) and no protein (CT) in 50 mM Tris pH 7.5, 200 mM KCl. The conductances are the mean (± SD) of the conductances measured over 10 min. (**B**) The capacitance of the t-LBM before and after incorporation of FupA/B (2 independent wells: red squares and green triangles) or buffer (blue diamonds). The times of protein addition and washes are indicated. Fifteen data points were used for each measurement. (**C**) The conductance after NaCl addition in the upper compartment of the t-LBM after incorporation of FupA/B (2 independent wells: red squares and green triangles) or buffer (blue diamonds). The conductances are the mean of the conductances measured during NaCl incubation (10 min for each NaCl concentration). (**D**) Immunoblot analysis using the anti-FupA antibody of FupA/B solubilized in the t-LBM well by SDS (lane T). Marker molecular weights are indicated (kDa).

**Figure 4 ijms-21-05496-f004:**
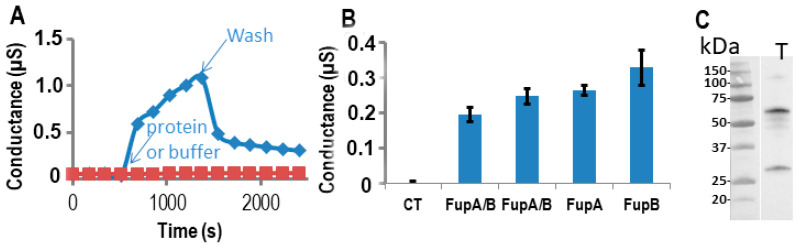
Measurements of Fup protein porin activity using impedance spectrometry. The proteins were digested by the TEV enzyme in the well. (**A**) The conductance of the t-LBM after the insertion of FupA/B (blue diamonds) and without protein (red squares). Time of protein (or buffer) addition and washes are indicated. (**B**) Conductance of the t-LBM (± SD) after the insertion of FupA or FupB compared to FupA/B (in two independent wells), and no protein (CT). The conductances are the mean of the conductances measured over 10 min. (**C**) Immunoblot analysis of FupA solubilized in the t-LBM well by SDS and revealed by antibodies specific to FupA (laneT). Marker molecular weights are indicated (kDa).

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
