# Peer review of "Physicochemical Evidence that Francisella FupA and FupB Proteins Are Porins"

_ijms, 2020, doi:10.3390/ijms21155496_

Round 1

Reviewer 1 Report

This study shows construction of FupA/B proteoliposomes and functional characterization. The study has its importance in the field of Francisella infection where no potent vaccine is available.

Strengths-
Methods are described well with details
The MS is written in scientific manner with clarity.
Results support the conclusions

Weakness-
Limited experimental work and data shown.

Author Response

Thank you for your encouraging remarks.

Reviewer 2 stated that:

The authors are including too many “method” in the “Results and Discussions”

Response: This has been carefully taken into consideration and 4 points have been moved to the ‘method’ section.

We have also remowed all citations in the conclusion.

Reviewer 2 Report

While the authors have presented their work in detail, there are issues with the structure. The authors are including too many “method” in the “Results and Discussions”. In order to allow the readers to have a good follow-up, the methods should be moved to the appropriate section. This would also allow other researchers with reproducibility.

I do not understand why the authors have citations in the conclusion. The authors seem to be discussing their result in the conclusion. The conclusion should address the key findings and make recommendations is any.

In the statistical analysis, the authors need to say exactly what analysis were performed and for what parameters.

Author Response

Thank you for your encouraging remarks.

Remark 1: The authors are including too many “method” in the “Results and Discussions”

Response: This has been carefully taken into consideration and the following points have been moved to the ‘method’ section:

- use of Triton X100 for the proteoliposome formation

- quantification of the dotblot (Fig. 2B) using the ImageLab software (Biorad, USA)

- we chose to vary NaCl as KCl was already in the buffer A (so on both sides of the t-LBM).

- Following conductance measurements, the wells of the t-LBM device were thoroughly washed to eliminate any soluble as well as any protein adsorbed on the device’s wells. The proteins potentially inserted in the t-LBM were then solubilized by addition of a SDS solution into the wells, separated by SDS-PAGE, transferred to a nitrocellulose membrane and probed with anti-FupA/B specific antibodies.

Remark 2: I do not understand why the authors have citations in the conclusion.

Response: The citations in the conclusion have been removed. When needed these citations have been moved to the ‘introduction’ section.

Remark 3: In the statistical analysis, the authors need to say exactly what analysis were performed and for what parameters.

Response: In Fig. 2C, we are comparing 2 independent samples of equal size, therefore we have specified in the legend:

Non-parametric one-way ANOVA